# Validation of the Arabic Levels of Emotional Awareness Scale (LEAS-Arb)

**Maimounah Hebi[1], Johanna Czamanski-Cohen[1,2,3], Karen L. Weihs [ID][3]\*, Richard D. Lane[3]**

**1** School of Creative Arts Therapies, Faculty of Social Welfare and Health Sciences, University of Haifa, Haifa, Israel, **2** The Emili Sagol Creative Arts Therapies Research Center, Faculty of Social Welfare and Health Sciences, University of Haifa, Haifa, Israel, **3** Department of Psychiatry, College of Medicine, The University of Arizona, Tucson, Arizona, United States of America

\* weihs@email.arizona.edu

## Abstract

Emotional awareness is a cognitive skill that has gained recognition as being important for psychological and physical health. It is measured with the Levels of Emotional Awareness Scale (LEAS), which is a 20-item written performance measure in which an individual is asked to write about how they and another person would feel in a series of hypothetical scenarios. The LEAS has been translated and validated in 19 languages. The aim of the current paper is to describe the translation and validation of the LEAS in Arabic (LEAS-Arb). To achieve our goal, we recruited 134 Arabic-speaking adults ages 18 and above to complete the LEAS in Arabic, along with additional self-report and performance measures that have been used in previous LEAS validation studies. Our findings support the reliability and validity of the Arabic translation of the LEAS originally created in English. The interrater reliability between the two scorers for the first ten study participants was very high, Cronbach's $\alpha = 0.951$ for self, 0.997 for other, and 0.962 for total LEAS scores. Internal consistency of the 20 LEAS items alpha scores for LEAS-Arb were self-Cronbach's $\alpha = .81$, other $\alpha = .96$, and total $\alpha = .86$. In addition, LEAS-Arb scores demonstrated convergent validity through expected associations with related constructs, including positive mood and alexithymia, consistent with prior LEAS validation studies. This study can facilitate the use of LEAS-Arb in varied settings across different Arabic speaking populations, which will allow better understanding of awareness of emotion in the Arab world and cross-culturally. This easily accessible measure can be used in clinical and research settings. Further research is needed to examine cross-cultural differences in emotional awareness as a function of culture and language.

## Introduction

Emotional awareness is a cognitive skill defined as the ability to recognize and describe emotion in oneself and others. It is a separate line of cognitive development

**Data availability statement:** Because our data set includes open responses to questions with the potential of including sensitive and private information, through which participants may identify themselves or others based on the responses, we are not able to upload the data set to a repository. The data set is available to any reasonable inquiry on request to the ethics committee of the faculty of welfare and health sciences at the University of Haifa at: ethics-health@univ.haifa.ac.il. Furthermore, because the data were collected before putting datasets in repositories was customary, we did not include this in our consent form. Because of this it is ethically problematic to upload the data.

**Funding:** The author(s) received no specific funding for this work.

**Competing interests:** Richard D. Lane has disclosed an outside interest in the Electronic Levels of Emotional Awareness Scale to the University of Arizona. Conflicts of interest resulting from this interest are being managed by The University of Arizona in accordance with its policies.

**Abbreviation:** MSS: Mental State Stories; LEAS: Levels of Emotional Awareness scale; LEAS-Arb: Levels of Emotional Awareness Scale in Arabic; LEAS-C: Levels of Emotional Awareness Scale for Children; PANAS: The Positive and Negative Affect Schedule; PAT: Perception of Affect Task; TAS: The Toronto Alexithymia Scale.

independent from other cognition domains [1]. Emotional awareness is measured using the 20-item Levels of Emotional Awareness Scale (LEAS), a written performance index [2]. Subjects write about their anticipated feelings and those of another person in response to short vignettes that describe interpersonal situations.

The emotional awareness framework suggests a hierarchical model ranging from basic undifferentiated sensations to highly integrated emotional states [3]. The levels of emotional awareness in ascending order are awareness of physical sensations, action tendencies, single emotions, blends of emotion, and combinations of blends of emotional experience [1]. The levels of emotional awareness highlight traits such as emotional differentiation, descriptive ability, and empathy [4]. Thus, LEAS measures the trait level at which an individual typically functions and this can inform how unexpressed or unrecognized emotions might impact their mental and psychological well-being [1]. The LEAS evaluates emotional awareness by interpreting written responses, so language structure and proficiency play a role in determining outcomes.

The reliability and validity of the LEAS have been established in normative and clinical contexts [5]. Lane et al. [2] initially validated the scale in 40 undergraduate students (20 men) and found it to correlate positively with openness to experience and emotional range—thus providing support for the LEAS as a valid cognitive–structural emotion construct.

Human beings exhibit a remarkable capacity for abstraction and introspection, with notable variability in emotional awareness across individuals [3]. Research to date shows that emotional awareness is associated with gender, education level, developmental stage/age (reading levels peak between ages 25–30) and socio-economic status (one's basic needs should be met to facilitate reflection on emotions) [6]. In addition to the above-mentioned factors, language and culture may influence an individual's trait level of emotional awareness. However, in the original study the LEAS was not found to relate to specific emotions, repressive coping style, or number of words used to describe emotion. Thus, it measures variation in levels of emotional awareness rather than measuring the quality or intensity of the emotion [7]. The LEAS measures emotional awareness across different cognitive levels, capturing both implicit and explicit emotional processes ($N = 2524$ [8]). An experimental study ($N = 390$) found that higher alexithymia (difficulty describing emotion in words) and lower emotional awareness scores correlated with lower accuracy in emotion recognition [9]. There are now 230 papers on the LEAS, demonstrating its relevance in preclinical and clinical contexts and its importance as a construct in the understanding of human health and well-being [5].

Emotional awareness is increasingly conceptualized as a trainable cognitive skill, rather than a fixed trait. Recent empirical evidence demonstrates that emotional awareness can be significantly improved through structured emotional skills training, with concurrent gains observed in related capacities such as interoceptive awareness, emotion regulation, and interpersonal emotion management [10]. The LEAS is uniquely suited to this framework, as it enables assessment of baseline emotional awareness and is sensitive to changes in skill level following intervention. Importantly,

assessment of emotional awareness is more accurate when conducted in an individual's native language, particularly among multilingual populations. Given that Arabic is spoken by hundreds of millions of people worldwide, the availability of a validated Arabic version of the LEAS has substantial relevance for both cross-cultural research and clinical practice, extending the application of this widely used instrument to Arabic-speaking populations.

Because the LEAS relies on written language to express differentiated emotions, accurate assessment depends on administering the measure in the respondent's dominant language. APA guidance on psychological assessment emphasizes matching test language to the examinee's language dominance to protect validity and avoid compromised performance due to language proficiency [11]. This is consistent with prior LEAS validation work in other languages (e.g., German and French), which underscores the need for language-specific adaptation and psychometric evaluation rather than ad hoc translation [12,13].

In this study, we aimed to validate the LEAS in Arabic. Arabic is the official language of 22 countries, in addition to 1.2 million speakers in the United States and two million in France [14], making Arabic the mother tongue of more than 400 million speakers [15] and one of the world's most used languages [15]. The richness of the Arabic lexicon stems from its large vocabulary (over 12.3 million words) and its two types, standard (classical Arabic-Fusha- written) and vernacular (spoken, colloquial-Aamiya) [16]. Arabic has many words for similar emotions, for instance, about 60 words express *love* in Arabic [17–19]. The validation of the Arabic translation of the LEAS is essential to enable its use in future research on cross-cultural aspects of emotional awareness in Arab-speaking people.

## Methods

### Design

To validate the Arabic version of the LEAS (LEAS-Arb) we translated the LEAS scale from English into the classical language "Fusha" (فصحى), which all Arabic readers and writers can understand. The LEAS Arabic validation was conducted similarly to those for previous validations of the LEAS in German ($N = 331$; [20]), French ($N = 121$; [21]), Japanese ($N = 344$; [22]), and Portuguese ($N = 176$; [23]), as well as for children ([LEAS-C] $N = 51$; [24]), by correlating LEAS scores with related psychological constructs. We administered the LEAS first to avoid priming effects from other inventories. We recruited participants through a company that specializes in providing participants for research (Midgam; Israel) ($N = 70$) and we recruited bachelor's degree students ($N = 112$), who were required to participate in studies for course credit as part of an Introduction to Psychology class of the Faculty of Welfare and Health Sciences at the University of Haifa. Inclusion criteria were age over 18 and fluent in reading and writing Arabic. The validation procedure for the LEAS-Arb followed the methodological framework described in Lane et al. and subsequent validation studies [2]. The Ethics Committee at the University of Haifa approved the study (approval number 003/17). Participants recruitment was conducted from August 5, 2017, to May 20, 2018. Informed written consent was obtained from all participants prior to any data collection.

### Procedure

**Sample.** Our study population was comprised of 182 adults (49 men, 121 women) who completed online questionnaires. Twenty-seven participants data could not be used because they did not complete at least 17 of the 20 items of the LEAS. Of the remaining 155 individuals, 21 participants were excluded from the data analyses because their Perception of Affect Task (PAT) scores were three standard deviations below the mean of the sample, indicating their failure to pass the attention check [25]. Thus, the final sample comprised 134 participants (see Fig 1).

### Measures

**LEAS.** The LEAS is a written performance measure that asks respondents to describe their anticipated feelings and those of another person in response to each of 20 vignettes described in two to four sentences, leading to three separate

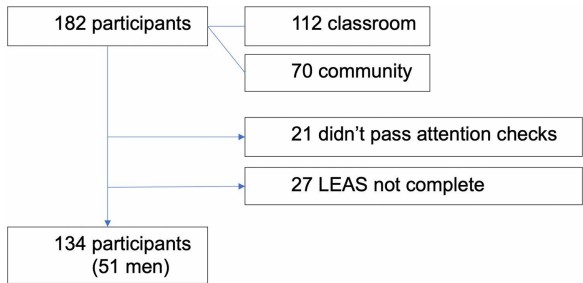

**Fig 1. Flow chart of participant recruitment and participation.**

LEAS scores: Self, Other and Total for each vignette and aggregated across the protocol. The LEAS scoring involves evaluating written responses for emotional content, with scores ranging from non-emotional to highly specific emotional terms, as outlined in the *LEAS Scoring Manual* [26], with scores from 0 to 3 indicating levels: 0 (*non-emotion word*), 1 (*words for physical reactions*), 2 (*general undifferentiated terms used in an emotional context*, e.g., "good" or "bad"), and 3 (*specific emotion word*) [3]. Inter-rater reliability of total scores on the LEAS between two independent raters was reported to be 0.81 in two studies [2,20], German version). The test–retest reliability of the LEAS after 2 weeks was.67.

A translator who was a native Arabic speaker with mother tongue English language proficiency translated the LEAS scenarios and questions from English to Arabic, then they were back translated to Arabic by another translator. Both versions were compared to ensure the accuracy of the translations. The translations were then provided to a convenience sample of 10 community members who confirmed that the translation was easy to understand.

Before scoring the LEAS data from study participants, two scorers were trained by the first and second authors using the scoring manual written by the fourth author. They practiced scoring by rating responses on LEAS items from the practice section of the scoring manual. After coding each word in the responses from 0 to 3, raters calculated scores for responses attributed to the self and the other person in the scenario. If at least two Level-3 words are found in a response, it receives a score of 4. If all the words that score 3 are redundant, then the self or other score is 3. Thus, self and other scores vary from 0 to 4, indicating increasingly differentiated emotional awareness and calculated by using the highest word score in the response for the self and for the other. Total scores are calculated based upon the self and other scores and range from 0 to 5 for the scenario. If no more than one of the self and other scores is 4, then the total score corresponds to the higher of the two scores. A total score of 5 is assigned only when both the self and other responses receive a score of 4 and the emotional content of the two responses is clearly distinct. If both the self and other responses receive a score of 4 but rely on identical or redundant emotion words, the total score is 4 rather than 5. Finally, scores across all scenarios are summed to yield overall self, other, and total LEAS scores (see Fig 2 for an example of LEAS scoring).

Rater training included rating the words from LEAS practice scenarios. They discussed their ratings until accurate scores were achieved and congruence was obtained. The raters demonstrated fidelity to the gold standard ratings of other LEAS responses provided in the training manual by independently scoring 20 scales, from which an interclass correlation coefficient of 0.962 between the two raters was obtained [26].

**PAT.** The PAT is a 140-item instrument that asks subjects to identify emotions in four 35-item subtasks [27]. In each subtask, five sets of stimuli targeting each of seven emotions (happiness, sadness, fear, anger, surprise, disgust, and neutral) are presented in Arabic translation that we translated. The four subtasks consist of verbal (sentences) or nonverbal (faces) stimuli with either verbal (words) or non-verbal (faces or pictures of scenes without faces) responses, permitting the following stimulus-response combinations: verbal/verbal, verbal/non-verbal, non-verbal/verbal and non-verbal/non-verbal.

"You and your best friend are in the same line of work. There is a prize given annually to the best performance of the year. The two of you work hard to win the prize. One night the winner is announced: Your friend. How would you feel? How would your friend feel?"

Answers varying by level of emotional awareness:

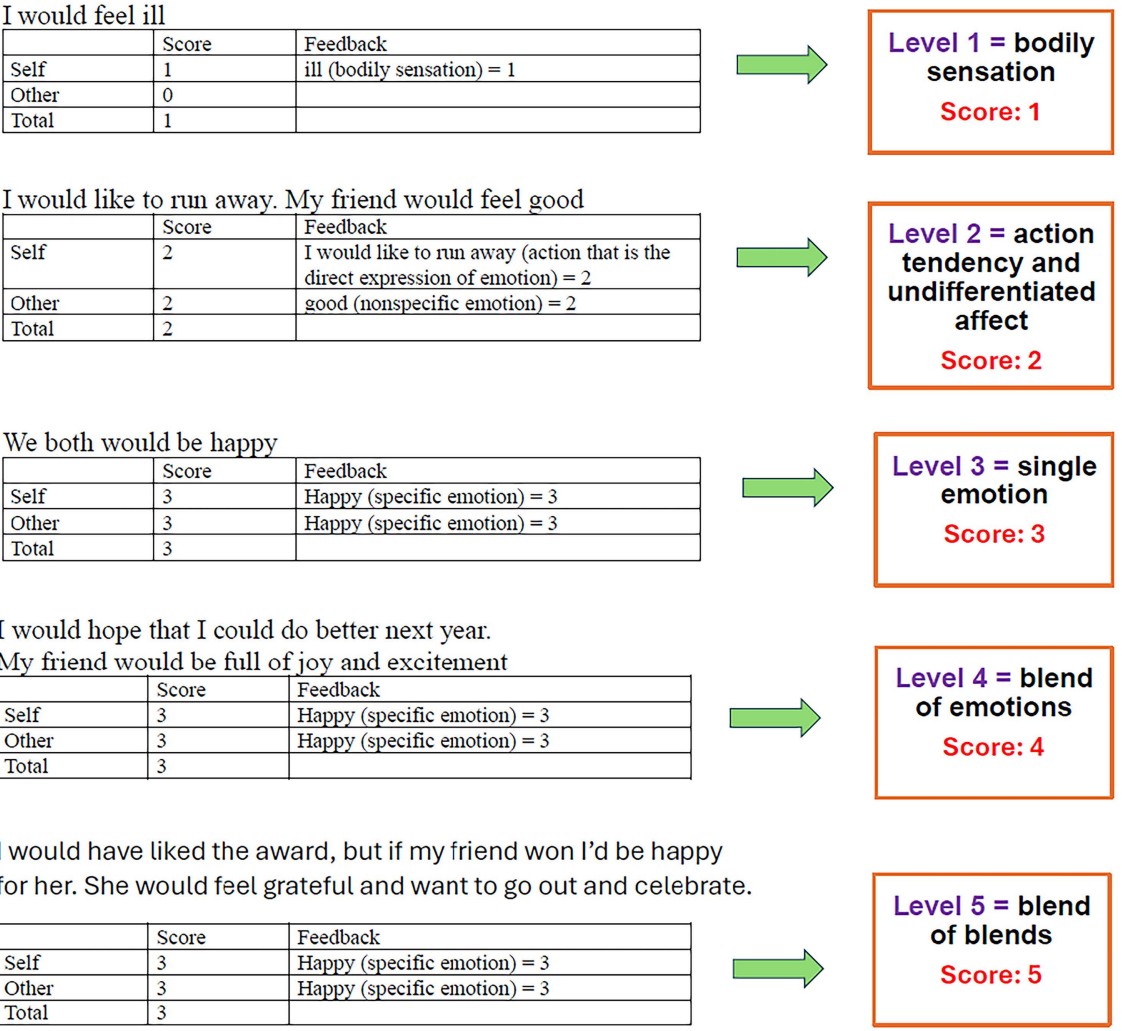

I would feel ill

|  | Score | Feedback |
|---|---|---|
| Self | 1 | ill (bodily sensation) = 1 |
| Other | 0 |  |
| Total | 1 |  |

Level 1 = bodily sensation

Score: 1

I would like to run away. My friend would feel good

|  | Score | Feedback |
|---|---|---|
| Self | 2 | I would like to run away (action that is the direct expression of emotion) = 2 |
| Other | 2 | good (nonspecific emotion) = 2 |
| Total | 2 |  |

Level 2 = action tendency and undifferentiated affect

Score: 2

We both would be happy

|  | Score | Feedback |
|---|---|---|
| Self | 3 | Happy (specific emotion) = 3 |
| Other | 3 | Happy (specific emotion) = 3 |
| Total | 3 |  |

Level 3 = single emotion

Score: 3

I would hope that I could do better next year.
My friend would be full of joy and excitement

|  | Score | Feedback |
|---|---|---|
| Self | 3 | Happy (specific emotion) = 3 |
| Other | 3 | Happy (specific emotion) = 3 |
| Total | 3 |  |

Level 4 = blend of emotions

Score: 4

I would have liked the award, but if my friend won I'd be happy for her. She would feel grateful and want to go out and celebrate.

|  | Score | Feedback |
|---|---|---|
| Self | 3 | Happy (specific emotion) = 3 |
| Other | 3 | Happy (specific emotion) = 3 |
| Total | 3 |  |

Level 5 = blend of blends

Score: 5

**Fig 2. Example LEAS Scoring (Item 20).**

Respondents match from a display of seven response options corresponding to the seven emotions. Scores consist of the proportion of accurate matches with a maximum score of 1.0. The total for Cronbach's alpha in our sample was .87. The mean accuracy of the task (mean percentage of correct answers) was 82.13%.

**Object relations inventory.** Participants are provided 5 minutes to describe a parental figure (in our case Mother) in writing (this instruction was translated into Arabic by the first author and back translated into English by a native Arabic and English speaker). Their essays are rated on a continuum from the lowest level, providing concrete descriptions without self–other differentiation, to the highest level of complexity, which includes differentiated and complex psychological characteristics that portray the individual as unique [28]. Rater training for the Object Relations Inventory

was conducted as described above for the LEAS to obtain inter-rater reliability of 0.918 prior to raters generating ratings for actual study data.

**Mental State Stories.** Mental state stories (MSS) is a measure of cognitive theory of mind function that requires true–false answers to questions about simple stories involving people and objects [29]. In this study, we used the subtasks involving inferences about people that include 12 scenarios of the 48 in the original MSS, which we translated to Arabic, and back translated from Arabic to English by different people. Both versions were compared to ensure the accuracy of the translations; some were deemed unsuitable to the Arab cultural context and were changed to comparable scenarios. For example, we replaced the word *pie*, which would be uncommon in Arabic, with the word *cake*. Higher scores reflect a greater ability to make mental or physical state inferences. In our study, Cronbach's alpha was.71.

**20-Item toronto alexithymia scale.** The Toronto Alexithymia Scale (TAS-20) is a 20-item self-report scale assessing emotional and social competency impairments. Respondents rate each item on a 5-point Likert scale for a maximum score of 100. The 20 items are divided into three subscales: difficulty identifying feelings, difficulty describing feelings, and externally-oriented thinking. Although alexithymia is a dimensional construct, cutoff scores have been established permitting the categorization of respondents into alexithymic (> 61), intermediate (52–60), or nonalexithymic (< 51) groups [30]. We adapted a validated Arabic version of the TAS-20 [31] to our study ($\alpha = .84$, N = 368). In our sample, $\alpha = .79$.

**Positive affect and negative affect schedule.** The Positive and Negative Affect Schedule (PANAS) are two 10-item mood scales [32]. Respondents rated the extent to which they are now experiencing each of 20 emotions on a 5-point Likert Scale ranging from *very slightly* to *very much*. A validated Arabic version of PANAS was used [33]. In our study, the Cronbach's alpha was.83.

**Demographics.** The participants reported their age, gender, marital status, education, self-rated income (above, below, or average income), ethnicity (Arab, Jew, or other), religion (Muslim, Christian, Druze, or other), and religiosity (secular, traditional, religious, or other). The SES was divided into a hierarchy of high, middle, and low SES based on the self-report of income level.

## Statistical analysis

All statistical analyses were conducted with SPSS version 25. Statistical significance was set at $p < 0.05$, two-tailed. Individual analyses of each rater's scores assessed interrater reliability for the scores of *self*, *other*, and *total*. To assess reliability of the LEAS in Arabic, Cronbach's alpha coefficients were calculated for the *self*, *other*, and *total* responses to the scenarios. Correlation coefficients were calculated for the mean LEAS scores (*self*, *other*, and *total*), TAS-20 (*total* and each *factor*), PAT, PANAS, and MSS. Correlations between the LEAS and other measures were conducted. *T* tests were used for comparisons between men and women participants, and a one-factor analysis of variance for the comparisons based on socioeconomic status and education. We also compared mean LEAS scores with a comparable U.S. English-speaking population as well as an undergraduate sample from Yale University, using a *t* test. In addition, we did a weighted means analysis to account for the differences in proportions of men vs women in the current sample as compared to the U.S English speaking samples.

## Results

### Reliability of LEAS-Arb

We observed excellent interrater reliability for the LEAS-Arb scores, consistent with prior validations [2], with Cronbach's $\alpha = 0.951$ for self, 0.997 for other, and 0.962 for total LEAS scores. Internal consistency of the 20 LEAS items alpha scores for LEAS-Arb were *self*-Cronbach's $\alpha = .81$, *other* $\alpha = .96$, and total $\alpha = .86$.

### Differences in LEAS-Arb scores by gender, education and Socioeconomic status (SES)

Of the 134 study participants, 38% were men and 62% were women. The mean age was 30.1 years (*SD* = 9.26; range 18–64); 78% were under 35 years; 48% were married, and 46% single. The rest were divorced or missing data. Most (72%)

had a bachelor's degree or were current university students, 69% were employed, and 56% had an average income (based on self- report). The sample was comprised of 71% Muslims, 12% Christians, and 15% Druzes (See Table 1).

In our sample, women scored significantly higher on the LEAS-Arb ($M = 68.4$, $SD = 11.9$) than men ($M = 62.3$, $SD = 13.2$), $t(131) = 2.76$, $p < 0.05$. Emotional awareness was statistically significantly related to being employed ($r = .246$, $p < .05$). Mean LEAS-Arb scores were higher in individuals with higher education (BA, MA, PhD; $M = 70.9$, $SD = 7.2$) vs high school education or less ($M = 60.6$, $SD = 12.8$), $t(147) = 4.48$, $p = 0.0001$, and with higher SES ($M = 68.6$, $SD = 12.7$) versus lower SES ($M = 62.6$, $SD = 13.3$), $F(5,122) = 4.086$, $p < 0.05$, $\eta^2 = 0.51$. Scores did not differ by age.

### Relationship between LEAS-Arb and other psychological variables

Associations between the mean scores of LEAS-Arb (self, other and total) and the related psychological constructs (the PAT, Object Relations Inventory and MSS, TAS-20 and its subscales, and PANAS, were assessed. The LEAS-Arb total scores were correlated with the PAT ($r = .306**$, $p \le .005$), Object Relations Inventory ($r = .290**$, $p \le .005$) and the MSS ($r = .473**$, $p \le .005$) and positive mood ($r = .299**$, $p \le .005$). TAS-20 ($r = -.013$) and negative affect ($r = .055$) scores were not significantly correlated with LEAS-Arb. Please see Table 2 for associations between the self and other LEAS-Arb scores and other constructs.

### Comparison between LEAS-Arabic Scores and LEAS-English Scores

The mean scores of total LEAS-Arb in Israel ($M = 66.04$, $SD = 12.7$) were significantly higher than those reported in English in the United States ($M = 61.9$, $SD = 10.7$), $t(518) = 3.66$, $p = 0.0003$ [9], but lower than the Yale University sample

**Table 1. Demographic data.**

|  |  | N | % |
|---|---|---|---|
| **Gender** | Male | 53 | 39.00% |
|  | Female | 81 | 61.00% |
| **Marital status** | Married | 65 | 48.50% |
|  | Divorced/widowed | 4 | 3.00% |
|  | Single | 62 | 46.30% |
|  | Other | 3 | 2.20% |
| **Education** | Elementary | 16 | 11.90% |
|  | High School | 96 | 71.60% |
|  | Graduate (MA) | 15 | 11.20% |
|  | Graduate (PhD) | 1 | 0.70% |
|  | Other | 6 | 4.50% |
| **Employment** | Yes | 92 | 69.00% |
|  | No | 42 | 31% |
| **Income** | Below average | 38 | 28.40% |
|  | Average | 75 | 56.00% |
|  | Above average | 21 | 15.70% |
| **Religion** | Muslim | 95 | 70.90% |
|  | Christian | 17 | 12.70% |
|  | Druze | 21 | 15.70% |
|  | Other | 1 | 0.70% |

NOTE: N = number of study participants

% = proportion of participants within each demographic category

**Table 2. Mean (Standard Deviation) and ranges of psychological constructs and their relationship with LEAS-Arb self, other and total scores.**

| Variable | M (SD) | Range | LEAS self | LEAS other | LEAS total |
|---|---|---|---|---|---|
| LEAS self | 58.29 (+9.81) | 21-78 | | | |
| LEAS other | 39.62 (+23.66) | 20-70 | | | |
| LEAS total | 66.04 (+12.76) | 24-90 | | | |
| Perception of affect task | 82.13 (+12.2) | 34-100 | | | |
| Blatt's parental description | 5.67 (+1.93) | 1-9 | .296** | .313** | .290** |
| Mental state stories | 20.27 (+3.75) | 4-24 | .442** | .423** | .473** |
| TAS-20 alexithymia general | 53.99 (+10.66) | 24-84 | .005 | .020 | −.013 |
| TAS-20 Difficulty describing feelings | 14.37 (+3.71) | 5-24 | .115 | .086 | .089 |
| TAS-20 Difficulty identifying feelings | 19.09 (+6.21) | 7-35 | .113 | .050 | .070 |
| TAS-20 Externally oriented thinking | 20.48 (+3.90) | 8-30 | −.221* | −.159 | −.212* |
| PANAS positive emotion | 38.08 (+6.62) | 19-50 | .258** | .282** | .299** |
| PANAS negative emotion | 24.64 (+7.45) | 11-42 | .055 | .092 | .055 |

NOTE: LEAS-Arb = Arabic Levels of Emotional Awareness Scale

TAS-20 = Toronto Alexithymia Scale, 20 items

PANAS = Positive and Negative Affect Scale

* $p \leq .05$; ** $p \leq .01$

(M = 72.1, SD = 8.2), t(225)=4.35, p < 0.001. Because most of our sample was comprised of women (n = 81), we conducted a weighted means analysis and compared our weighted mean (M = 65.4, SD = 12.9) with the LEAS-English scores (M = 61.9, SD = 10.7), t (518) = 3.07, p = 0.002. Since the LEAS is traditionally higher in higher educated individuals, we examined the scores of individuals in our sample who are students (n = 112) (M = 67.09, SD = 12.1) with the Yale University student sample (n = 95) (M = 72.1, SD = 8.2), and found that our highly educated participants had lower LEAS scores, t(205) = 3.44, p = 0.001.

## Discussion

Our findings support the reliability and validity of the Arabic translation of the original English LEAS, indicating the Arabic translation can be used to assess the emotional awareness level of Arabic-speakers. Our preliminary analysis showed excellent interrater reliability for scoring the LEAS-Arb. Construct and concurrent validity were demonstrated by positive correlations between LEAS-Arb and affective perception task (PAT), complexity and self-other different ion when describing one's parents (Object Relations Inventory), and theory of mind (MSS).

The correlation of the Externally Oriented Thinking (EOT) subscale of the TAS-20 with LEAS self and total scores in this study is also consistent with recent expert opinion regarding the association of these two measures [34]. A recent meta-analysis of 21 studies that used the LEAS and TAS-20 in the same participants revealed a low but statistically significant correlation between them, of r = −0.12 [35]. EOT has the strongest negative correlation with LEAS of the three TAS-20 subscales and this result reflects the prerequisite of ability to focus on one's internal thinking to detect one's emotions. In addition, like other studies, LEAS-Arb scores were not associated with negative affect. Thus, we conclude that the Arabic version of the LEAS is valid, as it is for English and other languages evaluated to date. The limited convergent validity observed between the LEAS-Arb and the TAS-20 warrants careful consideration. In the present study, significant associations emerged only with the Externally Oriented Thinking (EOT) subscale of the TAS-20, while the other two subscales were not significantly related to LEAS scores. This pattern is consistent with findings from a large-scale study of English-speaking participants (n = 380), in which LEAS scores were similarly associated with EOT but not

with the remaining TAS-20 factors [36], suggesting that this result represents a replication rather than a sample-specific limitation. The theoretical meaning of this selective association remains an open question. Recent models of alexithymia propose that the EOT factor may reflect deficits in interoceptive perception rather than a purely cognitive style [37], a conceptualization that aligns closely with the core construct assessed by the LEAS. More broadly, the modest magnitude of associations between the TAS-20 and the LEAS across studies likely reflects fundamental differences in both method and content, as the TAS-20 is a self-report measure whereas the LEAS is a performance-based assessment of emotional awareness [34]. Taken together, these considerations suggest that the observed pattern of correlations does not undermine the convergent validity of the LEAS-Arb but highlights the complementary nature of these instruments within the broader alexithymia spectrum.

An interesting finding in this study is a positive relationship between LEAS-Arb scores and positive emotion. In a Japanese sample the LEAS correlated positively with extraversion, which is associated with positive affect [38]. Both the current Arabic sample and the Japanese sample are derived from collectivist cultures. An association between LEAS and positive affect or extraversion may reflect the higher emphasis on maintenance and expression of positive affect in collectivist cultures, where one's impact on the group has higher priority than in individualistic cultures.

Consistent with other LEAS validation studies, our study also demonstrates higher scores in women and those with higher education level [39]. Among the 134 (81 female) participants in this study, women had higher scores than men on the LEAS-Arb, consistent with the original LEAS, which lends support to the cross-cultural consistency of this gender difference.

The mean scores of total LEAS in our sample were significantly higher than those in the U.S., French, and Japanese populations. The high scores may relate to the Arabic language being emotion-rich, offering more ways to express emotions [19,40], than languages in Europe, the United States, and Japan [15]. In Japan, which also has a collectivist culture, the LEAS scores were lower than in Western samples, while ours were higher. Given that Western culture is individualistic and individuals are encouraged to express their inner states and feelings [41,42] whereas Easterners tend to adjust to the group to maintain societal harmony, this study raises fascinating questions about under what circumstances a collectivist culture scores lower on emotional awareness and in what circumstances it does not do so. The privacy of writing responses on the LEAS-Arb provides participants with a secure medium for emotional expression, minimally influenced by societal constrains [3].

Our study recognizes that in cultures with Eastern-collectivist identity—traditional Arab societies—people learn to keep their feelings to themselves and prioritize the other and the community [43]. Because of that, one could have expected that *other* scores might be higher than *self* scores; however, our results did not show a higher *other* mean score on the LEAS-Arb. One explanation is that we translated the LEAS to classical/literary Arabic, and most participants answered and wrote in the same classical language. The use of literary language may have contributed to participants feeling less constrained by societal expectations and more connected to their inner selves because they were using the language of literature and poetry. In addition, as mentioned above the Arabic language abounds with forms of assertion and exaggeration and has more meanings for each word, indicating different emotional levels [44]. This means the participants' use of a rich language may have allowed them to express their emotional awareness in a creative and comfortable way. The LEAS has been orally administered and can work in samples with poor reading and writing skills [45]. Future studies could examine whether results are similar if participants were asked to answer in the spoken colloquial Arabic, and not the classic written language.

Future research using the LEAS-Arb should examine its relationship with both ability-based and trait-based models of emotional intelligence. Conceptually, emotional awareness as assessed by the LEAS aligns closely with the ability model of emotional intelligence, particularly the emotion perception branch, and prior findings suggest stronger associations with performance-based measures such as the MSCEIT than with self-report trait measures [46,47] Recent work further suggests that emotional awareness may represent a foundational cognitive mechanism supporting higher-order emotional intelligence abilities, including emotion regulation [48,49].

Weaknesses of this validation study also need to be considered. We had difficulty recruiting participants from diverse backgrounds resulting in limited demographic heterogeneity. In addition, a substantial proportion of the participants we recruited online did not complete the LEAS or pass attention checks. Thus, it is difficult to determine whether those who reliably filled out the questionaire were simply more inclined to express emotion.

A further limitation of the present study is that the factorial structure of the LEAS-Arb was not examined. Factor analytic procedures were not undertaken due to sample size considerations and are planned for future research once a larger and more statistically adequate sample is available. Importantly, previous factor analytic work on the LEAS in English has consistently supported a single-factor structure [50], and to date there is no evidence from validation studies in other languages suggesting a multidimensional solution. Nevertheless, examining the factor structure of the Arabic version remains an important step to confirm structural consistency across languages.

Furthermore, there is an inherent limitation in the LEAS dependence on written language skills, although social interaction factors may potentially complicate oral assessments.

These results underscore the utility of LEAS-Arb across Arabic speaking population, promoting deeper insights into emotional awareness within and beyond cultural context. This easily accessible measure can be used in clinical and research settings.

## Conclusions

The LEAS-Arb is a reliable and valid method for assessing emotional awareness in Arabic speaking individuals. Availability of the LEAS-Arb is a tool for the study of emotional awareness in Arabic speaking samples with diverse populations, in different clinical and nonclinical populations, as well as in older and less educated Arab adults. Arab culture is itself quite diverse and future research is needed to examine emotional awareness in different Arabic-speaking subgroups.

## Acknowledgments

The authors would like to thank the participants of this study, as well as to those who assisted in recruiting them.

## Author contributions

**Conceptualization:** Maimounah Hebi, Johanna Czamanski-Cohen, Karen L Weihs, Richard D Lane.

**Data curation:** Maimounah Hebi, Johanna Czamanski-Cohen.

**Formal analysis:** Maimounah Hebi.

**Methodology:** Richard D Lane.

**Project administration:** Maimounah Hebi, Johanna Czamanski-Cohen.

**Supervision:** Johanna Czamanski-Cohen, Karen L Weihs, Richard D Lane.

**Visualization:** Maimounah Hebi.

**Writing – original draft:** Maimounah Hebi.

**Writing – review & editing:** Johanna Czamanski-Cohen, Karen L Weihs, Richard D Lane.

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
