## [Decision Letter · Decision Letter 0]

30 Dec 2025

Dear Dr. Weihs,

The manuscript is a good piece of work that could potentially be published with a few modifications. I consider the requested changes to be minor but necessary to ensure the manuscript is of the highest quality. Please respond to the reviewers' requests. If you consider any of them to be inappropriate, please explain why.

We look forward to receiving your revised manuscript.

Kind regards,

Juan-Carlos Pérez-González, Ph.D.

Academic Editor

PLOS One

“Richard D. Lane has disclosed an outside interest in the Electronic Levels of Emotional Awareness Scale to the University of Arizona.  Conflicts of interest resulting from this interest are being managed by The University of Arizona in accordance with its policies.”

Reviewers' comments:

Reviewer's Responses to Questions

**Comments to the Author**

1. Is the manuscript technically sound, and do the data support the conclusions?

Reviewer #1: Yes

Reviewer #2: Yes

2. Has the statistical analysis been performed appropriately and rigorously?

Reviewer #1: Yes

Reviewer #2: Yes

3. Have the authors made all data underlying the findings in their manuscript fully available?

Reviewer #1: No

Reviewer #2: Yes

4. Is the manuscript presented in an intelligible fashion and written in standard English?

Reviewer #1: Yes

Reviewer #2: Yes

Reviewer #1: The paper presents a convincing and rigorous Arabic adaptation of the renowned LEAS. My recommendation for improvement is fourfold, but simple to implement:

a) The abstract should include a mention of the convergent validity of the instrument with respect to alexithymia and positive mood. The adaptation cannot be defended solely based on the internal consistency indices.

b) The discussion should critically analyse the limited evidence of convergent validity with respect to alexithymia, given that significant correlations have only been found with the Externally Oriented Thinking component.

c) Add to the limitations the fact that the factorial structure has not been explored. It could be added that this has been postponed for a later study once a larger and sufficient sample size has been achieved.

d) Among the future lines of research with the LEAS, it would be advisable to explore its relationship with both ability emotional intelligence (Mayer et al., 2016; Emotion Review) and trait emotional intelligence (Petrides et al., 2016; Emotion Review), given that these constructs have captured growing and dominant attention in the last two decades.

Reviewer #2: Overall, the thesis topic is of great significance, the writing format is standardized, and the research results are true and reliable. However, there are several minor issues as follows:

The introduction needs to elaborate on the significance of the LEAS and its application in other countries.

When giving examples of LEAS scoring, please verify the case of the full score of 5 points. Only when both the self and other dimensions are scored 4 points can the total score be counted as 5 points.

**Do you want your identity to be public for this peer review?** For information about this choice, including consent withdrawal, please see our For information about this choice, including consent withdrawal, please see our Privacy Policy .

Reviewer #1: No

Reviewer #2: **Yes:** Haibin WangHaibin Wang

---

## [Author Response · Author response to Decision Letter 1]

14 Feb 2026

Dear Editor and Reviewers,

We would like to thank the Editor and Reviewers for their careful evaluation of our manuscript and for the constructive and helpful comments. We have revised the manuscript accordingly and believe that these changes have substantially improved its clarity, rigor, and presentation. Below, we provide a point-by-point response to all comments. All changes in the manuscript are indicated in the revised version.

1. We revised the manuscript and associated files to ensure full compliance with PLOS ONE style and file-naming requirements, following the journal’s formatting sample guidelines.

2. We completed the PLOS ONE Inclusivity in Global Research questionnaire and have uploaded it as Supporting Information with the revised submission.

3. We updated the Competing Interests statement in the cover letter to explicitly confirm that this disclosed interest does not alter our adherence to PLOS ONE policies on sharing data and materials. No additional restrictions apply beyond those described in the Data Availability Statement.

4. We revised the Data Availability Statement to clarify that public deposition of the full dataset would breach compliance with the protocol approved by the University of Haifa Ethics Committee due to the sensitive nature of the human participant data. Accordingly, we request an ethics-based exemption and specify that deidentified data are available through a controlled access process, as detailed in the revised statement.

Reviewer #1, Comment (a):

The abstract should include a mention of the convergent validity of the instrument with respect to alexithymia and positive mood. The adaptation cannot be defended solely based on the internal consistency indices.

We revised the abstract to explicitly report evidence of convergent validity, including theoretically consistent associations with alexithymia and positive mood, in addition to internal consistency indices.

Reviewer #1, Comment (b):

The discussion should critically analyse the limited evidence of convergent validity with respect to alexithymia, given that significant correlations have only been found with the Externally Oriented Thinking component.

We expanded the Discussion section to critically address the limited convergent validity observed with alexithymia. Specifically, we clarified that the selective association with the Externally Oriented Thinking subscale replicates prior large-scale findings and discussed theoretical and methodological explanations for this pattern, including differences between self-report and performance-based measures.

Reviewer #1, Comment (c):

Add to the limitations the fact that the factorial structure has not been explored. It could be added that this has been postponed for a later study once a larger and sufficient sample size has been achieved.

We added this point to the Limitations section, noting that the factorial structure of the LEAS-Arb was not examined due to sample size considerations and that factor analytic procedures are planned for future research once a larger and more statistically adequate sample is available.

Reviewer #2, Comment:

The introduction needs to elaborate on the significance of the LEAS and its application in other countries.

We expanded the Introduction to elaborate on the significance of the LEAS and its international applications. Specifically, we highlighted evidence that emotional awareness is a trainable skill, the suitability of the LEAS for assessing change following intervention, and the importance of administering the instrument in respondents’ native language, with reference to prior language-specific validations in other countries.

Reviewer #2, Comment:

When giving examples of LEAS scoring, please verify the case of the full score of 5 points. Only when both the self and other dimensions are scored 4 points can the total score be counted as 5 points.

We clarified the LEAS scoring procedure in the Methods section to specify that a total score of 5 is assigned only when both self and other responses receive a score of 4 and contain distinct, non-redundant emotional content; otherwise, the total score is 4. This clarification appears in the paragraph describing LEAS scoring, immediately preceding the reference to Figure 2.

---

## [Editor Report · Decision Letter 1]

16 Feb 2026

Validation of The Arabic Levels of Emotional Awareness Scale (LEAS-Arb)

PONE-D-25-45274R1

Dear Dr. Weihs,

We’re pleased to inform you that your manuscript has been judged scientifically suitable for publication and will be formally accepted for publication once it meets all outstanding technical requirements.

Kind regards,

Juan-Carlos Pérez-González, Ph.D.

Academic Editor

PLOS One

---

## [Editor Report · Acceptance letter]

PONE-D-25-45274R1

PLOS One

Dear Dr. Weihs,

I'm pleased to inform you that your manuscript has been deemed suitable for publication in PLOS One. Congratulations! Your manuscript is now being handed over to our production team.

Kind regards,

on behalf of

Dr. Juan-Carlos Pérez-González

Academic Editor

PLOS One